# Cytokine-Induced Killer (CIK) Cells, In Vitro Expanded under Good Manufacturing Process (GMP) Conditions, Remain Stable over Time after Cryopreservation

**DOI:** 10.3390/ph13050093

**Published:** 2020-05-12

**Authors:** Katia Mareschi, Aloe Adamini, Sara Castiglia, Deborah Rustichelli, Laura Castello, Alessandra Mandese, Marco Leone, Giuseppe Pinnetta, Giulia Mesiano, Ivana Ferrero, Franca Fagioli

**Affiliations:** 1Department of Public Health and Paediatrics, The University of Turin, piazza Polonia 94. 10126 Torino, Italy; mleone@cittadellasalute.to.it (M.L.); franca.fagioli@unito.it (F.F.); 2Stem Cell Transplantation and Cellular Therapy Laboratory, Paediatric Onco-Haematology Division, Regina Margherita Children’s Hospital, City of Health and Science of Turin; 10126 Torino, Italy; aadamini@cittadellasalute.to.it (A.A.); scastiglia@cittadellasalute.to.it (S.C.); drustichelli@cittadellasalute.to.it (D.R.); lauramarisa.castello@gmail.com (L.C.); amandese@cittadellasalute.to.it (A.M.); gpinnetta@cittadellasalute.to.it (G.P.); iferrero@cittadellasalute.to.it (I.F.); 3Department of Oncology, University of Turin, 10060 Torino, Italy; giulia.mesiano@ircc.it; 4Candiolo Cancer Institute, FPO—IRCCS—Str. Prov.le 142, km. 3,95, 10060 Candiolo (TO), Italy

**Keywords:** advanced therapy medicinal product (ATMP), good manufacturing practice (GMP), cytokine-induced killer (CIK) cells, cryopreservation, drug formulation, stability program

## Abstract

Cytokine-induced killer (CIK) cells are advanced therapy medicinal products, so their production and freezing process has to be validated before their clinical use, to verify their stability as a drug formulation according to the good manufacturing practice (GMP) guidelines. We designed a stability program for our GMP-manufactured CIK cells, evaluating the viability, identity and potency of cryopreserved CIK cells at varying time periods from freezing, and compared them with fresh CIK cells. We evaluated the effects of the cryopreservation method, transportation, and the length of time of different process phases (pre-freezing, freezing and post-thawing) on the stability of CIK cells. This included a worst case for each stage. The expanded CIK cells were viable for up to 30 min from the addition of the freezing solution, when transported on dry ice within 48 h once frozen, within 60 min from thawing and from 12 months of freezing while preserving their cytotoxic effects. The reference samples, cryopreserved simultaneously in tubes and following the same method, were considered representative of the batch and useful in the case of further analysis. Data obtained from this drug stability program can inform the accurate use of CIK cells in clinical settings.

## 1. Introduction

In recent times, the research and development of drug formulations in which the main biological action is carried out by cells or tissue has led to the definition of a new category of medicines: advanced therapy medicinal products (ATMP). The use of living cells in the final product formulation imposes some limitations on their use because of their rapid loss of viability and, consequently, their function after cryopreservation. Their short shelf-life in the final product formulation means it is necessary to validate a suitable cryopreservation method to maintain the functional characteristics of those cells over time. Following a good manufacturing practice (GMP) production process, cryopreservation allows for the preservation of the cells in quarantine until all the sterility tests required to ensure their safety are completed. Moreover, in the case of multiple dose infusion, this allows for all the scheduled doses of a drug formulation.

In recent decades, several protocols of cellular therapy using cytokine-induced killer (CIK) cells [1] were established for their experimental clinical use. The GMP standard culture conditions for the production of CIK cells have already been successfully validated by expanding in vitro peripheral blood mononuclear cells (PBMCs) after an initial priming with interferon-*γ* and anti-CD3 antibody, followed by repeated stimulation with interleukin-2 [2,3,4]. These T cells with an NK phenotype, characterized by a very high cytolytic potential, displayed potent cytotoxic activity in hematological and solid tumors in autologous and allogeneic settings in the presence of very limited toxicity [5,6,7,8].

In our cell factory, we validated CIK cell production under GMP conditions by cultivating PBMCs in standard conditions for 3 weeks of expansion [9] to use them in a phase I experimental protocol for patients with relapsed sarcomas. At the end of their production, the cells were frozen in bags to allow for dose escalation in the Phase I clinical trial. For the ATMP, the freezing process had to be validated to verify the stability of the cells (drug formulation), following GMP production for medicine products guidelines [10].

The cryopreservation process consisted of freezing the cells at very low temperatures by adding cryoprotective agents to avoid damage to the cells caused by the formation of ice crystals. The sample was then progressively cooled until it reached the temperature of nitrogen vapors and was stored for a precise time period, thus prolonging its expiration date. However, freezing/thawing cycles could affect cell stability and lead to lower cellular viability, recovery and function [11,12].

On these bases, we designed an in-house stability program with manufactured CIK cells and evaluated the viability, identity and potency of cryopreserved CIK cells at different stages from the point of freezing, and then compared them with fresh CIK cells. The operational thawing flowchart and study design can be seen in Figure 1. Through this study, we validated the cryopreservation process, transportation from the production site to the cryogenic room, and thence on to the clinical unit. The thawing method and post-thawing stability tests of CIK cells were also performed by following GMP conditions in the cell factory at the Paediatric Onco-Haematology Division, City of Health and Science University Hospital of Turin.

## 2. Results

### 2.1. CIK Cell Expansion

The CIK cells were obtained by cultivating peripheral blood mononuclear cells (PBMCs) as previously mentioned in [9]. The CIK cells were harvested after 21–23 days of culture, and data regarding their cellular growth and viability during expansion is illustrated in the Appendix A, respectively. Sequential CIK cell applications with escalating cell doses were manufactured according to ATMP regulations for clinical application. We expanded four batches and obtained good cellular expansion for all four with a median fold increase of 22.7 (range 21.1–92.2) for the CD3 population. Each quality control (QC) test (sterility, viability, identity) at release on these products was compliant with the defined release criteria (see Appendix A) with the exception of Batch 2, which failed to comply with viability at 67%. Although this viability did not meet the acceptance criteria (>80%), Batch 2 was cryopreserved and not excluded from the study, so we could validate the stability program over time in terms of post-thawing viability, identity and potency.

Batches 1, 3 and 4 were analyzed for the freezing process, transportation and stability program prior to freezing and post thawing.

### 2.2. Validation of the Cryopreservation Method and Transportation Process before Freezing

To validate the cryopreservation method of CIK cells, we tested the cells’ viability, identity and the time period in which the cells remained stable after adding the freezing solution, containing DMSO (see Figure 1A). We also included worst cases where cells remained in the freezing solution until 60 min before starting the freezing process in the controlled rate freezer. We predicted the conditions that had the potential to activate worst cases could include (1) technical issues in the cell factory or (2) transportation if freezing was not immediate. We matched the viability and the identity of CIK cells frozen for different time periods to those of a fresh drug formulation in pre-cryopreservation conditions. The difference was indicated in terms of Δ%. At the time of thawing, all the bags were intact and did not exhibit any damage. The readability of the identification labels, the temperature and times during transportation from the cell factory to the cryogenic room each resulted as compliant (data not shown).

#### 2.2.1. Evaluation of Viability

Apart from Batch 4, where viability largely remained unchanged, the viability remained stable for Batches 1, 2 and 3 only within 30 min from the addition of the freezing solution before the start of the cryopreservation process, while after 60 min this decreased and caused two cases of out-of-specification (OOS) with a Δ% of viability of >15% (see Figure 2). The values of viability were, however, greater than 70% in both cases. There were no statistically significant differences in the observed parameters.

#### 2.2.2. Evaluation of Identity

The testing conditions were all compliant in terms of their identity since the Δ% of CD45 + CD3 + and the Δ% of CD45 + CD3 + CD56+ from the pre-freezing time period were <10%, as shown in Table 1. Some values of Δ% displayed a negative sign because the percentage values of CD45 + CD3+ and/or CD45 + CD3 + CD56+ in post-thawing conditions were slightly greater than the reference one (pre-freezing). This was due to higher CIK cell concentrations in the post-thawing condition compared to the pre-thawing one, since the cytofluorimetric method excluded the dead cells during the morphological analysis.

### 2.3. Validation of Transportation Process before Thawing

To validate the transportation process (see Figure 1B) and test the stability of the drug formulation, the bags containing CIK cells were removed from the nitrogen vapors, and the viability of the thawed cells was compared to the cells that had been kept on dry ice for 24 and then 48 h, before thawing. During thawing, all the bags were intact and did not display any damage. The readability of the identification labels, the cell transportation on dry ice from the cryogenic room to the QC laboratory for the thawing process and the recorded temperature were all compliant (data not shown). Each batch of CIK cells was compliant in terms of viability after 24 h and after 48 h (Δ% < 15%), as shown in Figure 3.

### 2.4. Stability Program of the Drug Formulation over Time and after Thawing

In order to verify the stability of the drug formulation after various time periods of storage in nitrogen vapors, for up to 6 months and a maximum of 1 year, we evaluated its identity, viability and cytotoxicity by comparing the manufactured pre-freezing CIK cells vs. thawed CIK cells at various post-thawing time periods and considering a Δ% < 20 compliant for each analysis.

Moreover, we evaluated the stability of the drug formulation after thawing to establish a time limit before thawing, up to which the viability of the CIK cells remained compliant.

After 3 weeks of expansion, the cells were frozen in bags (from five to 10 depending on the final number of CIK cells) at a concentration of 50 × 10^6^ cells/mL. For each bag, three tubes were also cryopreserved to be used as reference samples for the tests during validation.

As previously reported, Batch 2 was not compliant because its viability was less than 80%. Although the value did not meet the acceptance criteria, Batch 2 was nevertheless cryopreserved and not excluded from the study, so we were able to validate the transportation, freezing process and stability program. Batch 2 was evaluated for stability, identity and potency at the time of thawing to evaluate it for the possible use of those cells in a specific clinical context.

Viability was tested for each batch at different stages in the bags and tubes. Each thawed bag at the different stages showed a viability compliant in terms of variability during pre-freezing and thawing. The bag thawed at 15 days post freezing was also evaluated for the viability of CIK cells after 30, 60 and 120 min from thawing, to evaluate it for the time limit in which the CIK cells remain viable before infusion into the patient.

#### 2.4.1. Evaluation of Viability

The mean viability evaluated over time in Batch 1 was 99.6 ± 0.6% pre-freezing and then 92.8 ± 1.7%, 88.4 ± 1.7%, 80.3 ± 3.1%, 82.5 ± 5.2% and 87.2 ± 5.7%, respectively, after 15 days, 1, 6, 9 and 12 months. In Batch 2 and Batch 3, the viability was evaluated after 15 days, 1, 6 and 9 months. The values of Batch 2 were 67.0 ± 1.5% pre-freezing and then 65.0 ± 4.3%, 68.0 ± 0.5%, 56.0 ± 5.1% and 65.2 ± 0.8%. The values obtained for Batch 3 were 91.9 ± 3.4% pre-freezing and then 80.6 ± 1.9%, 81.1 ± 8.1%, 80.4 ± 2.5% and 88.7 ± 3.4%. For Batch 4 the viability was evaluated up to 6 months and the values were 80.3 ± 2.9% pre-freezing and then 80.7 ± 1.7%, 81.0 ± 2.7%, 77.0 ± 1.02%, after 15 days, 1 month and 6 months, respectively. The Δ% for each analyzed condition, in comparison with the fresh product, was always <20%, as indicated in Figure 4, where all batches up to 6 months are reported. Additionally, in this case no statistical differences were observed.

The viability evaluated after 30 min from thawing in all batches was stable and decreased slightly after 60 min and significantly after 120 min, as shown in Figure 5. On the basis of these results, the thawed CIK cells could be infused up to 60 min from thawing.

#### 2.4.2. Evaluation of Identity

We also evaluated pre- and post-thawing identity for up to 6 months, as shown in Table 2, the Δ% from the pre-freezing condition was <10% in all the analyzed batches at the different time periods, thus compliant to the pre-established acceptance criteria. All the cryopreserved products remained stable in terms of cellular identity for up 9 and 12 months (data not shown).

#### 2.4.3. Evaluation of Cytotoxicity

To evaluate if the thawed cells preserved their cytotoxic activity, CIK cells, at different months post freezing, were co-cultured with sarcoma tumor cells (target cells) in different ratios. To ensure the same tumor target, it was decided to carry out the cytotoxicity test simultaneously on the cryopreserved CIK cells of all the batches available on the test date, corresponding to 6 months from the freezing of the first batch, as showed in Figure 6. In a second subsequent test, cytotoxicity was also assessed at 9 (*n* = 2) and 12 months (*n* = 1).

All the tested CIK cells showed a cytotoxic effect proportional to the target ratio and preserved their effect or function up to 6 months from cryopreservation. The batches analyzed at 9 and 12 months post freezing also showed a preservation of cytotoxic function. Batch 2, despite not being compliant with the release criteria in terms of viability, showed a cytotoxic action comparable to those with a higher percentage of CD45 + CD3 + CD56 +. The presence of NK cells was <10%, therefore we observed cytotoxic effects against sarcoma tumors related to the action of CIK cells.

#### 2.4.4. Evaluation of Reference Samples

To demonstrate that the cells frozen in the tubes represented the drug product, as defined in the ANNEX 19 [13], we analyzed the cell viability over time in the tubes in comparison with the bags. We observed a Δ% of < 10% from the pre-freezing condition in all analyzed samples, as shown in Figure 7.

## 3. Discussion

Numerous experimental clinical trials have been carried out over the last two decades, confirming the safety and therapeutic action of CIK cells in cancer patients [5,6,7,8,14,15,16].

The antineoplastic activity of CIK cells has also been studied in sarcomas and other solid tumors. In vitro and in vivo preclinical studies were conducted to evaluate the antitumor action performed by CIK cells isolated from patients diagnosed with sarcomas. Experimental models have been developed in an autologous setting, studying primary tumor cell lines from patients, with the aim of verifying the intrinsic variability of each tumor and individual patient [17,18]. Indeed, it has been shown that autologous CIK cells are able to induce apoptosis in the most aggressive tumor cells implicated in neoplastic recurrence, which are responsible for metastasis and resistance to chemotherapy.

Several phase I–II clinical trials have been developed over the past decade, with phase III and IV studies also underway to explore the safety, activity and effectiveness of CIK cell treatment in cohorts of patients with hematologic and solid malignancies [7,19,20,21].

A register of all preclinical and clinical studies with CIK cells has recently been established and data from the register [5,8] show that around 1800 cancer patients were newly treated at the time of the drafting of this document. The dosage of CIK cells used in the experimental clinical protocols is widely variable, with values ranging in absolute numbers from a few million to tens of billions, or in cell/kg values, ranging from 1 × 10^5^ to 12.4 × 10^9^/kg administered with a single multiple treatment cycle. The number of infusions each patient received is also extremely variable, but no serious side effects have been reported [5,7,8].

Our cell factory (AIFA authorization aM-147/2019) was accredited to produce and expand CIK cells under GMP conditions, by validating the best culture conditions and assessing the stability tests for both the raw material and the cytokines used [9].

The clinical protocol involves the expansion of CIK cells from lymphocytapheresis or autologous bloodletting. Since it is not considered ethically acceptable to subject patients to an invasive procedure such as lymphocytapheresis for a validation procedure, whole blood from voluntary donations of healthy subjects was used.

The purpose of this paper was to summarize the validation study results obtained on (1) cryopreservation and the thawing process of ATMP CIK cells (2) the transportation of the cells to identify the correct acceptance criteria to be applied, so as to evaluate the stability of the drug formulation and give useful information to clinicians for their subsequent administration to the patient.

In light of the results obtained, the processes of cryopreservation, transportation and thawing can be considered validated in defining the following acceptance criteria:(1)The expanded CIK cells remained viable within 30 min from the addition of the freezing solution (physiological solution + 5% human albumin + 10% DMSO) before the start of the cryopreservation process. After this time, we observed two OOS in two batches, therefore we considered 30 min the limit time within which the freezing process must start. This is an important consideration if the storage cryopreservation room is not as close to the cell factory production room as possible in a hospital/academic cell factory.(2)The CIK cells remained stable in terms of viability when kept on dry ice for up to 24 h and 48 h, since Δ% compared to the pre-freezing condition was less than 15%. Therefore, the transportation and permanence of the CIK cells on dry ice complies with the acceptance criteria. This means that, in the case of any clinical issues occurring with the patient, the thawing of CIK cells can be postponed for up to 48 h (while maintaining the cells on dry ice). This transportation condition can be considered a worst case in the case of no availability of a dry shipper with nitrogen vapors.(3)The CIK cells remained stable in terms of viability within 60 min from their thawing and significantly lost viability after 1 hour from their thawing. This means that the cells can be infused into patients within 1 hour from thawing without the quality of the product being altered.(4)The CIK cells remained stable in terms of cell viability, identity and potency within 6 months of their freezing. Preliminary data showed they retained stability within 9 and 12 months (data not shown).(5)All the tested CIK cells maintained a proportional killing trend of the curve on the different E:T ratios and were active for up to 6 months from their cryopreservation. We also observed a preservation of cytotoxic function in the batches analyzed at 9 and 12 months post freezing (data not shown). In addition, Batch 2, which was considered not compliant at release for the viability test of <80%, showed a significant cytotoxic action against cancer cells after thawing. This behavior opens up the possibility of using a non-compliant batch, according to the guidelines on GMP specific to advanced therapy medicinal products §1.5. “Administration of out of specification products” (ref.), on the basis of the clinical context of the patient.(6)The reference samples, cryopreserved in tubes simultaneously and using the same method as the drug product, really represent the batch, and can be used in the case of further analysis.

To our knowledge, no manuscripts describing the drug stability of CIK cells, such as ATMP, have been published and we maintain that the data obtained in this paper provides us with accurate information for the use of CIK cells for clinical purposes.

The results presented in this paper confirmed the maintenance of CIK cell characteristics up to 1 year from freezing, which reflects not only the compliance of the CIK cells obtained during validation, but also the effectiveness of the freezing methods, transportation and the conservation of the cryopreserved units in nitrogen vapor tanks.

## 4. Materials and Methods

### 4.1. Study Design and Setting of the Study

The CIK cell production and expansion methods were described in Castiglia et al. 2018 [9]. This process validation was performed on four batches produced in GMP conditions.

At the end of expansion, the cells were cryopreserved in physiological solutions containing 5% human serum albumin (HSA) and 10% dimethyl sulfoxide (DMSO), and were then stored in nitrogen vapor for the stability over time studies. The use of HSA comes from its use as a long-established supplement of cell culture media, its use in cryopreservation and as a formulation buffer for stem cell therapies.

For each batch, three conditions were established to evaluate their compatibility with the number of cells obtained for the validation of the cryopreservation, the transportation process and to evaluate the effects of DMSO over time on cell viability and on the stability of the product before freezing (see Figure 1A):(1)T0: the bag was subject to the cryopreservation process immediately after adding the freezing solution containing DMSO.(2)T-30′: the bag was subject to the cryopreservation process 30 min after the addition of the freezing solution.(3)T-60′: the bag was subject to the cryopreservation process 60 min after the addition of the freezing solution.

Furthermore, the T0 condition was produced in triplicate to assess the viability of the cryopreserved bag before thawing after 24 h (T0-1) and 48 h (T0-2) on dry ice. These conditions were established to evaluate the stability of the drug product during transportation on dry ice after the nitrogen tank pick up and before infusion (see Figure 1B).

Five additional bags were then produced to evaluate the stability of the manufacturing formulation at 15 days, 1 month, 6 months, 9 months and 1 year after cryopreservation (see Figure 1C). Moreover, the stability of the thawed drug product was also evaluated after 30, 60 and 120 min from thawing in the cryopreserved bags for 15 days (see Figure 1D).

For each condition, the three reference samples were also stored in tubes: one was used as a reference sample to evaluate the viability and compare it with the viability of the cells in the bag. The second tube was used to carry out the cytotoxicity test and the third was stored as a backup reference sample for a possible re-test.

The following parameters were evaluated upon thawing:(1)Integrity of the cryopreserved units and readability of the identification labels on the bags.(2)Temperature and time periods during transportation from the cell factory to the cryogenic room, and from the cryogenic room to the cell factory.(3)Cellular identity of cells in the bags.(4)Cellular viability with trypan blue and Bürker chamber calculations on bags and tubes.(5)Cytotoxic activity against sarcoma cells of post-thawed cells.

### 4.2. Peripheral Blood Collection and Peripheral Blood Mononuclear Cell (PBMCs) Isolation

The cells used as starting material were isolated from whole blood (WB) units from voluntary donations from healthy subjects deemed unsuitable for internal procedures by The Center for Production and Validation of Haemocomponents (CPVH) of the City of Health and Science Hospital of Turin.

The compliance of all virology tests was verified on the collected WB before its use. The blood was stored at room temperature and kept in quarantine until the results of the virology tests were provided. 

After this, the PBMCs were separated using Histopaque 1.077 g/L (Sigma-Aldrich, St. Louis, MO, USA) and then cultured and expanded as previously described [9].

### 4.3. CIK Cell Expansion

PBMCs were cultured in a serum-free X-VIVO 15 medium with the addition of 1% of penicillin/streptomycin (P/S) (Lonza, Allendale, NJ, USA) and IFN-*γ* (Boehringer Ingelheim, Vienna, Austria) at a final concentration of 1000 U/mL (day 0) as previously described [9]. Briefly, 30 × 10^6^ of those cells were seeded in non-treated flasks in 15 mL of culture medium (Corning Incorporated, Corning, NY, USA) with vented cap. The flasks were maintained in a vertical position to ensure a better oxygenation of the cellular suspension, a reduction of the culturing medium distribution surface and an appropriate culturing medium volume to re-suspend the cells homogeneously.

The following day (day + 1), Proleukin, which is an IL-2 cytokine (Novartis, Origgio, VA, Italy), and CD3 Pure, which is an IgG anti-CD3 (OKT-3) (Miltenyi Biotec, Bergisch Gladbach, Germany), were added to a final concentration of 300 U/mL and 50 ng/mL, respectively.

Every 48 or 72 h, a cell count was performed to monitor their expansion and, on the basis of cell growth, fresh medium with IL-2 (300 U/mL) was added to obtain a cellular concentration of between 1 and 1.5 × 10^6^ cells/mL. The culture was carried out for approximately 21 ± 3 days and the QCs were performed. The release criteria are shown in Appendix A.

### 4.4. CIK Cell Cryopreservation

The cells were re-suspended in a physiological solution with 5% HSA (Albital 200 g/L, Kedrion Biopharma, LU, Italy) and 10% DMSO (AL.CHI.MI.A. S.R.L., Ponte San Nicolò, PD, Italy). The drug product was at a concentration of 50 × 10^6^ cells/mL.

The volume of the bags varied from a minimum of 10 mL to a maximum of 30 mL per bag. The units were cryopreserved in a controlled temperature freezer (ICE cube 14S, SY-LAB Geräte GmbH, Neupurkersdorf, Austria) using slow cooling rates (approx. 1 °C/min), and were then stored in nitrogen vapor tanks.

### 4.5. CIK Cell Thawing

We proceeded to thaw the bag or the tube combined with each condition, placing it in a thermostatic bath at 37 °C and diluting 1:3 with hydroxyethyl starch (HES), Voluven 6% (Fresenius Kabi Italia S.R.L., Isola della Scala, VR, Italy) to evaluate cellular viability and identity as shown in Figure 1.

### 4.6. Cell Count and Viability Analysis

The CIK cells were triple counted with an optical microscope, using a Bürker chamber calculation as indicated in the European Pharmacopoeia (Chapter 2.7.29) [20]. The mean number of cells counted was considered compliant when there were between 80 and 150 cells in 25 squares of the Bürker chamber. The cellular viability was evaluated with the trypan blue method (Sigma-Aldrich, St. Louis, MO, USA). The acceptance criteria of cell viability were the following: Δ% < 15% post-thaw bag vs. pre-thaw, Δ% < 10% thawed reference sample in the tube vs. bag, Δ% < 15% post-thaw T-30′ and T-60′ vs. post-thaw T0 bag, Δ% < 20% post-thaw bags at more times vs. pre-freezing, where Δ% was calculated as (value pre-thawing – value post-thawing)/ value pre-thawing × 100.

### 4.7. Cellular Identity Analysis

Flow cytometry was performed according to the European Pharmacopoeia (Chapter 2.7.24) using a Beckman Coulter NAVIOS (Beckman Coulter, Brea, CA, USA) [9]. Basal flow cytometry was performed at day 0 in order to evaluate the lymphocyte subpopulations on PBMCs after gradient stratification.

To initiate flow cytometry, monoclonal antibodies CYTO-STAT TetraCHROME CD45-FITC, CD56-RD1, CD19 ECD and CD3-PC5 (Beckman Coulter, Brea, CA, USA) were used for B lymphocytes and CYTO-STAT TetraCHROME CD45-FITC, CD4-RD1, CD8-ECD and CD3-PC5 (Beckman Coulter, Brea, CA, USA) for T lymphocytes. After 21 ± 3 days of expansion, flow cytometry with CD3-FITC, CD56-PE and CD45 KO (Beckman Coulter, Brea, CA, USA) was performed to evaluate CIK cell identity. As a negative control, cells were incubated without antibodies. Data were analyzed using Navios software (Vs. 1.2, Beckman Coulter, Krefeld, Germany). For data analysis, we designed the physical gate as [A]. From this gate, we obtained the CD45 + CD3+ and the CD45 + CD56+ cell populations. On the dot plot we gated the double positive CIK cell population CD45 + CD3 + CD56+. On the same dot plot, we also identified the population of NK cells, CD45 + CD56 + CD3 −. The acceptance criterion of cell identity was the following: Δ% < 10% post-thaw bag vs. pre-thaw [9].

### 4.8. Cytotoxicity Test

CIK tumor killing activity was evaluated using a primary tumor cell line as a target, obtained from a biopsy sample of a patient with undifferentiated pleomorphic sarcoma, as previously described in [22]. We previously confirmed the complete membrane expression of the HLA molecule as well as the NKG2D ligand’s expression. CIK and target tumor cells were not HLA-matched because we ex vivo expanded CIK from healthy subjects (paragraph 4.2 Materials and Methods).

The cytotoxicity tests were performed post-thawing on representative aliquots of frozen samples, using the CellTiter-Glo^®^ Luminescent Cell Viability Assay method (Promega Italia s.r.l, Milano, Italy), to quantify the viable and metabolically active cells by evaluating the concentration of ATP; the measurements were recorded with a GloMax 96 Microplate Luminometer (Promega). CIK and tumor cells were co-cultured at progressively decreasing effector:target (E:T) ratios (40:1, 20:1, 10:1, 5:1, 2.5:1, 1:1, 1:2 and 1:4) for 72 h in 200 μL of culture medium with Proleukin at a concentration of 300 U/mL at 37 °C and 5% CO_2_. Each sample ratio was plated in triplicate, and tumor cells in the absence of CIK effector cells were used as a control to establish their spontaneous mortality, while CIK cells alone were used as a control to measure the spontaneous ATP release.

The percentage of tumor-specific CIK cell lysis for each E:T ratio was calculated with the following formula:

Target mortality = 100 − [(100/target spontaneous ATP release) × (specific E:T ratio ATP release−CIK cells spontaneous ATP release)]. To ensure the same tumor target, it was decided to carry out the cytotoxicity test simultaneously on the cryopreserved CIK cells on all the batches available on the date of the test, corresponding to 6 months from the freezing of the first batch. In a second subsequent test, cytotoxicity was assessed at 9 and 12 months.

### 4.9. Statistical Analysis

Cell growth, viability, immunophenotype and cytotoxicity assay data were analyzed and plotted with the GraphPad program (version Prism 8.4). The data were represented as mean ± standard deviation. A Shapiro–Wilk test was used for the normality evaluation for a small number of samples and a Wilcoxon signed-rank test was used instead of a paired *t*-test because the data distribution was not normal.

## Figures and Tables

**Figure 1 pharmaceuticals-13-00093-f001:**
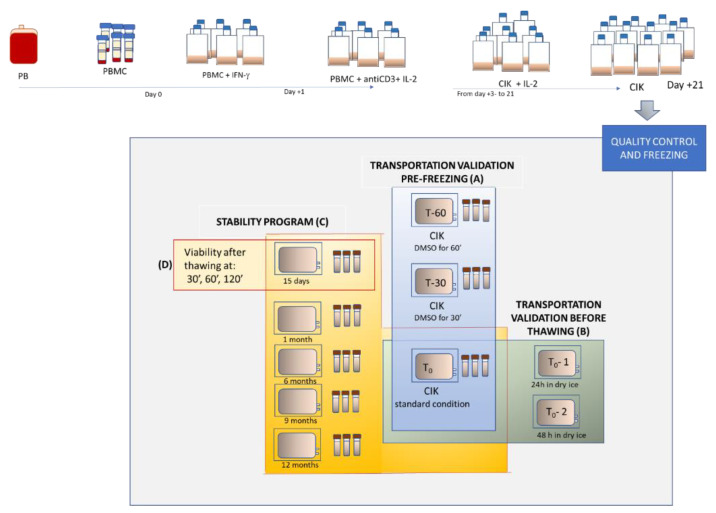
Study design scheme.

**Figure 2 pharmaceuticals-13-00093-f002:**
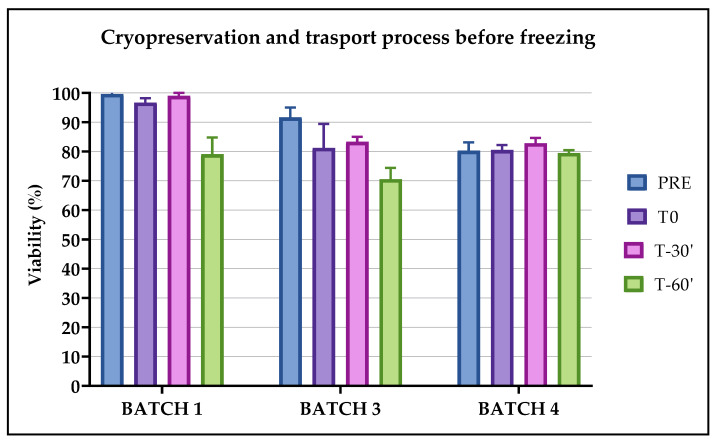
Viability on Batches 1, 3 and 4 evaluated for their validation of the cryopreservation and transportation processes before freezing. The percentage of viable cells was calculated on fresh product at the time of pre-freezing (PRE) and at different time periods after adding the freezing solution containing DMSO: immediately (T0), after 30 min (T-30′) and after 60 min (T-60′).

**Figure 3 pharmaceuticals-13-00093-f003:**
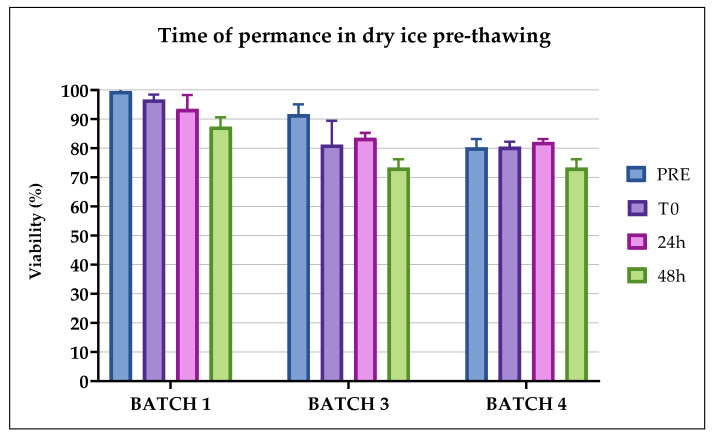
Viability on Batches 1, 3 and 4 evaluated for the validation of the cryopreservation and transportation processes before thawing. The percentage of viable cells was calculated for the CIK cells immediately after thawing, and then after 24 h and 48 h on dry ice, to validate the time for which CIK cells are stable prior to thawing.

**Figure 4 pharmaceuticals-13-00093-f004:**
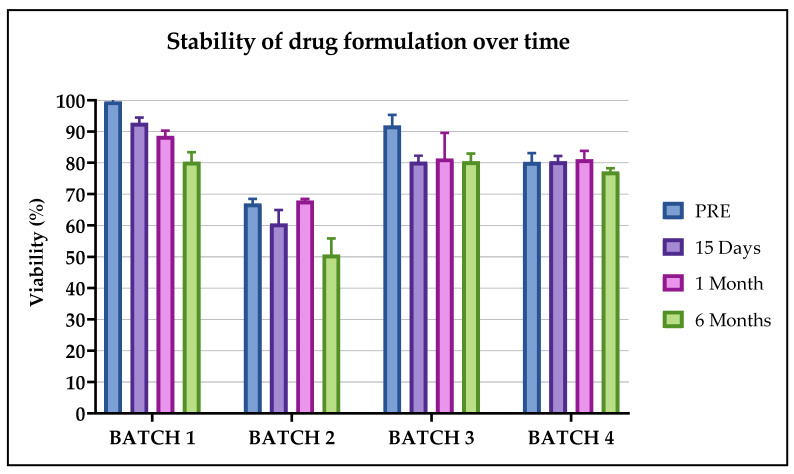
Viability of Batches 1, 2, 3 and 4 evaluated to validate the stability of the cellular product over time. The percentage of viable cells was calculated on fresh product at the time of pre-freezing (PRE) and at different time periods: 15 days, 1 month and 6 months.

**Figure 5 pharmaceuticals-13-00093-f005:**
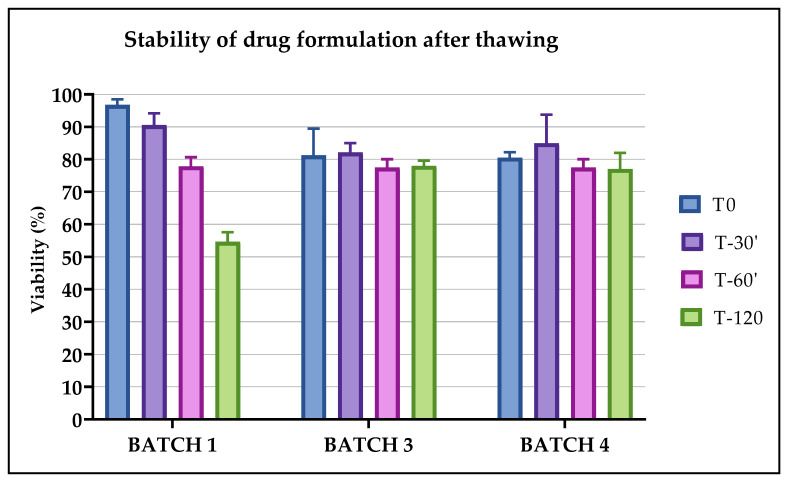
Viability of Batches 1, 2, 3 and 4 was evaluated to validate the stability of the cellular product after thawing. The percentage of viable cells was calculated for the bags thawed at 15 days post freezing (T0) and after 30 (T-30′), 60 (T-60′) and 120 (T-120′) min from the moment of thawing.

**Figure 6 pharmaceuticals-13-00093-f006:**
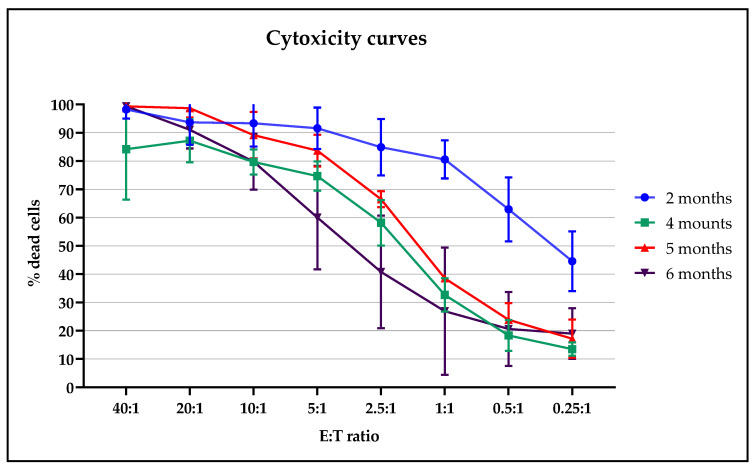
Cytotoxicity curves evaluated at various post-thawing time periods with scalar effector and target (E:T) ratios.

**Figure 7 pharmaceuticals-13-00093-f007:**
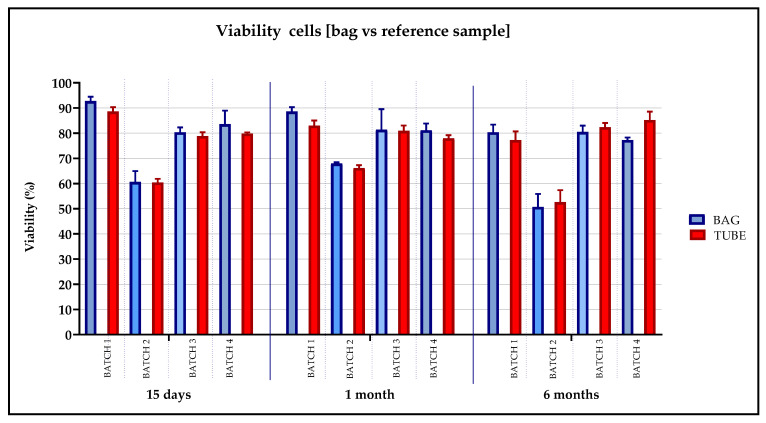
Viability evaluated on Batches 1, 2, 3 and 4 to validate the stability of the cellular product in bags and tubes. The percentage of viable cells was calculated concurrently on the bags (red histograms) and tubes (blue histograms) at the different cryopreservation time periods: 15 days, 1 month and 6 months.

**Table 1 pharmaceuticals-13-00093-t001:** Identity evaluated by immunophenotype analysis on Batches 1, 3 and 4. The values and relative Δ, in percentages, were evaluated between the immunophenotype of the post-thawing conditions and referred to as T0-1 (T0), T-30′ and T-60′ vs. pre-freezing conditions, referred to as PRE. CA = Acceptance criteria, the limits of which are indicated in bold, are reported. ND = condition not determined.

		% CD45 + 3 +	Δ% CD45 + 3 +	% CD45 + 3 + 56 +	Δ% CD45 + 3 + 56 +
	CA	>80%	<10%	≥15%	<10%
Batch 1	PRE	97.13		42.94	
	T0-1	96.46	0.70	40.57	5.52
	T-30’	97.17	−0.04	43.51	−1.33
	T-60’	96.13	1.03	42.82	0.28
Batch 3	PRE	98.70		52.00	
	T0-1	98.26	0.45	54.48	−4.77
	T-30’	98.75	−0.05	57.37	−10.30
	T-60’	98.70	0	54.25	−4.33
Batch 4	PRE	96.40		39.38	
	T0-1	93.61	2.90	36.83	6.48
	T-30’	96.95	−0.60	40.02	−1.63
	T-60’	96.32	0.10	41.29	−4.85

**Table 2 pharmaceuticals-13-00093-t002:** Identity evaluated by immunophenotype analysis on Batches 1, 2, 3 and 4. The values and relative Δ, in percentages, were evaluated between the immunophenotype of the post-thawing conditions at different time periods: 15 days, 1 month and 6 months, and in two cases, also at 9 and 12 months vs. pre-freezing conditions (PRE). CA = Acceptance, criteria, the limits of which are indicated in bold, are reported. ND = condition not determined.

		% CD45 + 3 +	Δ% CD45 + 3 +	% CD45 + 3 + 56 +	Δ% CD45 + 3 + 56 +
	CA	>80%	<10%	≥15%	<10%
Batch 1	PRE	97.13		42.94	
	15 days	96.55	0.60	42.41	1.23
	1 month	98.18	−1.10	44.50	−3.56
	6 months	95.76	1.41	41.04	4.42
Batch 2	PRE	90.60		50.00	
	15 days	91.68	−1.19	52.07	−4.14
	1 month	91.26	−0.73	51.55	−3.10
	6 months	90.77	−0.19	50.10	−0.20
Batch 3	PRE	98.70		52.00	
	15 days	97.98	0.73	56.93	−9.48
	1 month	97.42	1.30	61.91	−19.06
	6 months	98.14	0.57	55.77	−7.25
Batch 4	PRE	96.40		39.38	
	15 days	93.61	2.90	36.83	6.48
	1 month	95.44	0.99	37.62	4.47
	6 months	95.50	0.93	41.46	−5.28

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
