# Peer review of "Cytokine-Induced Killer (CIK) Cells, In Vitro Expanded under Good Manufacturing Process (GMP) Conditions, Remain Stable over Time after Cryopreservation"

_pharmaceuticals, 2020, doi:10.3390/ph13050093_

Round 1

Reviewer 1 Report

This paper describes protocol validation for the cryopreservation and recovery of Cytokine Induced Killer cells. The experimental procedures are well controlled, the methods are clearly described and the comparisons made are valid. The data are mostly presented appropriately

The following pints should be addressed.

  1. For each measurement (cell viability, cytotoxicity) how many replicates were tested on each cell batch/ experimental condition? This should be stated in the methods and figure legends and mean values for multiple replicates shown with standard deviations throughout the article.

  1. Statistical comparisons. There are no statistical tests presented to support the authors conclusions. The data presented should be analysed with the appropriate statistical comparisons between batches and between different experimental conditions.

  1. Cellular composition. What is the identity of the residual CD3- non-T cells after CIK generation? This is important because residual NK differences in the contribution of CD3-CD56+ NK cells could contribute to variation in the kill in cytotoxicity assay.

  1. The reference given for clinicaltrials.gov referring to the expansion protocol needs to be given more precisely to include the relevant trial identifier.

  1. Cytotoxicity assays: The authors mention that cells recovered at 2, 3, 5 and 6 months were assayed in a single batch, whereas those recovered after 6 and 9 months were assayed in a second batch. It is clear from the viability and cytotoxicity data that the assay variability between the two experiments cannot simply be represented in a single graph. I would suggest removing the 9 month and 12 month data from the graphs as the trend towards reduced viability and cytotoxic capacity is clear form the data up to 6 months of recovery. This applies to figure 4, 6 and 7. It seems that the 9 and 12 month data points are missing for some of the batch comparisons so I would recommend omitting these data. If necessary, the relevant for 9 and 12 months could be described in the text with emphasis that these were assayed in a separate experiment. The authors should also consider amending Table 2, either removing 9 and 12 month data or clearly marking that these timepoints were assayed separately.

  1. Cytotoxicity assays. More detail should be given for this protocol in the materials and methods, including the identity (Phenotypic characteristics including MHC class I expression) and origin of the sarcoma tumor cells used for these assays (are these in house or commonly used as standard target cells), and the length of assay incubation time. Were the CIK and the target cell HLA matched? If not, this is effectively an allogeneic kill model.

Minor points:

Supplementary figure 1. This legend is incomplete with the label for one batch missing

Author Response

Dear review

We thank you for the comments and observations.

We have completed our revision of the manuscript as you and other reviewers suggested. The changes to the first version are tracked in the manuscript in red. We have also added 2 references and modified the number of the references inserted by Zotero.

We hope that it will now be deemed suitable for publication in your Journal. Please do not hesitate to contact us with any further questions or requests for modification.

Yours sincerely,

Katia Mareschi

As far as your comments are concerned our responses are outlined as follows:

This paper describes protocol validation for the cryopreservation and recovery of Cytokine Induced Killer cells. The experimental procedures are well controlled, the methods are clearly described and the comparisons made are valid. The data are mostly presented appropriately

The following points should be addressed.

  1. For each measurement (cell viability, cytotoxicity) how many replicates were tested on each cell batch/ experimental condition? This should be stated in the methods and figure legends and mean values for multiple replicates shown with standard deviations throughout the article.

The measurements for cell viability and cytotoxicity were evaluated in triplicates and the values indicated in the graphics actually represent the mean values of the intra-experimental triplicate counts with standard deviation. For the cytotoxicity test, we also evaluated a triplicate from the same batch because we have 3 different tubes cryopreserved from the same batch; therefore in Figure 6 each value is the result of the average of three experiments, which as described in the methods section, were tripled.

In the reviewed manuscript, we reported in all figures the values with the standard deviations of each analysed condition

  1. Statistical comparisons. There are no statistical tests presented to support the authors conclusions. The data presented should be analysed with the appropriate statistical comparisons between batches and between different experimental conditions.

We added the paragraph about statistical analysis but since no statistical differences were observed in this validation study we based our null hypothesis on the fact that the values of cell viability and identity after thawing did not decrease more than 15-20% on the basis of our rationale; therefore we excluded the condition where a value was out of specification.

  1. Cellular composition. What is the identity of the residual CD3- non-T cells after CIK generation? This is important because residual NK differences in the contribution of CD3-CD56+ NK cells could contribute to variation in the kill in cytotoxicity assay.

We reported in paragraph 2.4.3 ‘Evaluation of cytotoxicity’ that the values of CD3-CD56+ NK resulted negligible because their presence were < 10%; therefore we retained that their anti-tumour cytotoxic effect was absolutely negligible for the cytotoxic test.

  1. The reference given for clinicaltrials.gov referring to the expansion protocol needs to be given more precisely to include the relevant trial identifier.

In clinicaltrials.gov we identified 81 clinical trials in which the isolation and expansion protocol of CIK cells, and the number of infusions for each patient received, are extremely variable. We added also a reference illustrating the report of the international registry on CIK cells (IRCC) where heterogeneity of the protocols has been discussed.

  1. Cytotoxicity assays: The authors mention that cells recovered at 2, 3, 5 and 6 months were assayed in a single batch, whereas those recovered after 6 and 9 months were assayed in a second batch. It is clear from the viability and cytotoxicity data that the assay variability between the two experiments cannot simply be represented in a single graph. I would suggest removing the 9 month and 12 month data from the graphs as the trend towards reduced viability and cytotoxic capacity is clear form the data up to 6 months of recovery. This applies to figure 4, 6 and 7. It seems that the 9 and 12 month data points are missing for some of the batch comparisons so I would recommend omitting these data. If necessary, the relevant for 9 and 12 months could be described in the text with emphasis that these were assayed in a separate experiment. The authors should also consider amending Table 2, either removing 9 and 12 month data or clearly marking that these timepoints were assayed separately.

Thank you for your comments., We have removed the 9- and 12-month data from Figures 4, 6 and 7 and in Table 2 and we modified the text.

  1. Cytotoxicity assays. More detail should be given for this protocol in the materials and methods, including the identity (Phenotypic characteristics including MHC class I expression) and origin of the sarcoma tumor cells used for these assays (are these in house or commonly used as standard target cells), and the length of assay incubation time. Were the CIK and the target cell HLA matched? If not, this is effectively an allogeneic kill model.

We added more details in paragraph 2.4.3 ‘Evaluation of cytotoxicity, from row 164 as suggested by the reviewer. To test the CIK tumor killing activity  we used a sarcoma line, as target obtained in house, from a biopsy sample of a patient with Undifferentiated Pleomorphic Sarcoma as described in the additional reference (Mesiano et al 2018). This was found to express the ligands of the NKG2D receptor mediator of CIK lysis and class I HLA molecules. In this work, for ethical issues, we isolated the CIK cells from the peripheral blood of healthy donors and cytotoxic tests were performed in an allogeneic setting. In our previous paper we compared activities in autologous and allogeneic settings and demonstrated that there were no significant differences, showing that the allogeneic component does not provide an advantage in terms of lysis to the CIK cells. The reason for this activity is based on the fact that the CIK cells kill tumour cells through a NKG2D-ligand mechanism and do not use TCR to kill. However, in our experimental clinical trial, CIK cells will be used in an autologous setting.

Minor points:

Supplementary figure 1. This legend is incomplete with the label for one batch missing

Thank you for your comment. This legend has been modified and corrected.

Reviewer 2 Report

This is an interesting manuscript on the GMP process of expanded CIK cells.

Major comments:

1. Number of CIK cells (range) frozen should be given.

2. The cell line used in figure 6 should be stated.

3. The original report on CIK cells by Schmidt-Wolf et al., J Exp Med 1991 should be referenced.

Author Response

Dear review

We thank you for the comments and observations.

We have completed our revision of the manuscript as you and other reviewers suggested. The changes to the first version are tracked in the manuscript in red. We hope that it will now be deemed suitable for publication in your Journal. Please do not hesitate to contact us with any further questions or requests for modification.

Yours sincerely,

Katia Mareschi

As far as your comments are concerned our responses are outlined as follows:

This is an interesting manuscript on the GMP process of expanded CIK cells.

Thank you for your comment

Major comments:

1. Number of CIK cells (range) frozen should be given.

The number of CIK cells obtained at the end of production and then frozen is described in the supplementary materials.

  1. The cell line used in figure 6 should be stated.

Details about the cytotoxic test have been described in the paragraph 2.4.3 ‘Evaluation of cytotoxicity’, from row 164.

  1. The original report on CIK cells by Schmidt-Wolf et al., J Exp Med 1991 should be referenced.

Thank you for your suggestion. We have now added this reference as reference N.1

We have also added another reference of the same author illustrating the report of the international registry on CIK cells and modified the number of the references inserted by Zotero.

Round 2

Reviewer 1 Report

Overall, the authors have addressed the previous comments.

Importantly,  attention will be needed to correct the English synatax and spelling throughout the manuscript but in particular figure titles and labels.

Author Response

Dear review,

Thank you for your attention. I’m sorry for the syntax errors which now are been corrected.

Moreover, an English native reviewer has further reviewed the final version of the paper.

The changes to the first version are tracked in the manuscript in red.

I hope that you will find the revised manuscript suitable for the publication in Pharmaceuticals.

Best regards

Yours sincerely,

Katia Mareschi

Reviewer 2 Report

None.

Author Response

(The authors gave the same response as above.)
